# Comparative transcriptomics and phylostratigraphy of Argentine ant odorant receptors

**Mathew A. Dittmann**[1]*, **Grzegorz Buczkowski**[1], **Michael Scharf**[2], **Brock A. Harpur**[1]

**1** Department of Entomology, Purdue University, West Lafayette, IN, United States of America, **2** University of Florida, Gainesville, FL, United States of America

\* madittmann1@gmail.com

**Data Availability Statement:** \*\*PA at Accept: Please follow up with AU for data sharing info\*\* Transcriptome data were submitted to NCBI under BioProject PRJNA1006049. (https://www.ncbi.nlm.nih.gov/bioproject/?term=PRJNA1006049) NCBI

## Abstract

Nestmate recognition in ants is regulated through the detection of cuticular hydrocarbons by odorant receptors (ORs) in the antennae. These ORs are crucial for maintaining colony cohesion that allows invasive ant species to dominate colonized environments. In the invasive Argentine ant, *Linepithema humile*, ORs regulating nestmate recognition are thought to be present in a clade of nine-exon odorant receptors, but the identity of the specific genes remains unknown. We sought to narrow down the list of candidate genes using transcriptomics and phylostratigraphy. Comparative transcriptomic analyses were conducted on the antennae, head, thorax, and legs of Argentine ant workers. We have identified a set of twenty-one nine-exon odorant receptors enriched in the antennae compared to the other tissues, allowing for downstream verification of whether they can detect Argentine ant cuticular hydrocarbons. Further investigation of these ORs could allow us to further understand the mechanisms underlying nestmate recognition and colony cohesion in ants.

## 1 Introduction

Eusocial insects are some of the most successful insect species on the planet, owing to the advantages afforded by their various social behaviors mediated through nestmate cooperation [1, 2]. This nestmate cooperation is based on the ability of individuals to distinguish nestmates from non-nestmates [3]. Colony members detect cuticular hydrocarbons (CHCs) present on the cuticle of individuals they encounter via antennation. During antennation, CHCs bind to odorant binding proteins, which shuttle them to odorant receptor (OR) complexes consisting of a tuning OR that binds to specific chemicals, which activates an OR co-receptor (*orco*) that opens an ion channel and activates the neuron [4]. These ORs are responsible for the regulation of all kinds of insect behavior, especially in eusocial insects where odor and pheromone cues regulate behaviors such as foraging, colony defense, and brood care [5–10]. However, because tuning ORs tend to detect specific chemicals, eusocial taxa such as ants that rely on odorant signaling typically have greatly expanded OR repertoires to manage the variety of pheromones required to maintain colony function [11–13]. This gene expansion has made it difficult to investigate how ORs regulate nestmate recognition and colony cohesion.

Accession numbers for the proteomes used to construct phylostratigraphy can be obtained in Figure S1. These Accession numbers can be searched on the NCBI Datasets site. (https://www.ncbi.nlm.nih.gov/datasets/) Results from phylostratigraphy analysis were stored on Dryad and are available from DOI: https://doi.org/10.5061/dryad.83bk3jb1d

**Funding:** The author(s) received no specific funding for this work.

**Competing interests:** The authors have declared that no competing interests exist.

Prior investigations into how ORs regulate behavior have typically been focused on manipulating the entire nestmate recognition system, rather than individual OR genes. Initial research was limited to understanding how CHCs influence nestmate aggression through manipulation of CHCs through environmental and dietary changes, or direct applications [14–17]. As genomic resources became available, the investigation of ORs themselves has recently become feasible [11, 18]. *orco* knockouts have been conducted in a couple of species, but remain impractical due to difficulties generating transgenic reproductives in most ant species [19, 20]. For those species where genetic manipulation is not feasible, chemical knockdown of *orco* is possible [21]. However, manipulating *orco* involves breaking the OR system, preventing investigation of the contributions of smaller groups of ORs. Functional characterization of individual ORs has been performed in some species, but remains difficult [22, 23]. While molecular tools such as RNAi or SYNCAS could potentially be used to investigate the contribution of individual tuning ORs to nestmate recognition, the expanded OR repertoire in ants makes choosing targets for investigation a difficult task [24, 25]. Before these studies can be conducted, prospective ORs that are likely associated with nestmate recognition must be identified. Given that invasive ants can be serious pests, especially species that form supercolonies such as the Argentine ant, investigating the mechanisms of nestmate recognition remains an important area of research.

Since the 20<sup>th</sup> century, the Argentine ant (*Linepithema humile*) has managed to expand from its native range in South America to invade every continent except Antarctica, becoming a major urban and agricultural pest in the process [26]. In areas it has invaded, the Argentine ant has managed to outcompete native ant species due to its tendency to form supercolonies [27]. Typically, ant colonies are aggressive towards conspecifics from different nests, but populations exhibiting unicolonial behavior do not exhibit this aggression, instead treating workers from other colonies as nestmates [28]. The loss of aggression towards other nests results in less resource expenditure due to intra-species competition, allowing Argentine ant supercolonies to concentrate colony resources towards outcompeting native ant species and other non-ant competitors [29–31]. The origins of this unicolonial behavior are primarily thought to have come from reductions in genetic diversity that occurred during invasion events, leading to a reduction of alleles controlling nestmate recognition, and ultimately resulting in the loss of intraspecific aggression [26, 32–34].

The ORs that mediate nestmate recognition in the Argentine ant have not yet been identified, and the large number present in the genome makes functional analysis of the ORs to determine their targets impractical [11]. To make investigation of nestmate recognition more approachable, we have identified potential ORs involved in sensing cuticular hydrocarbons. First, we generated tissue transcriptomes for the head, leg, thorax, and antennae in order to identify which OR genes showed increased expression in the antennae. Second, we performed a phylostratigraphy analysis to determine which *L. humile* ORs most recently evolved within the *L. humile* proteome. These analyses will narrow down the list of potential ORs most likely to be involved in nestmate recognition, laying the groundwork for future investigations into the origins of eusocial colony cohesion, as well as supercoloniality in highly invasive ant species.

## 2 Methods

### 2.1 Sample collection

Foraging workers were collected from *L. humile* laboratory colonies using an aspirator, suspended in RNALater (AM7021, ThermoFisher, Waltham, MA) solution, then chilled and stored at 4°C. Workers were separated into antennae, head, leg, and thorax tissues via

microscissor dissection, while the abdomen was discarded. For each body part, tissue from approximately 200–250 workers was collected and stored in separate RNALater tubes at -80˚C. Four biological replicates for each tissue were collected, and one replicate of antennae and leg tissues were discarded due to quality issues. Once all samples were collected, total mRNA was extracted using an SV Total RNA Isolation System kit (Z3101, Promega, Madison, WI). The resulting samples were sent to Novogene for transcriptome sequencing using Illumina Novaseq (80M pair-ended 150bp reads per sample). All sequence data are publicly available via NCBI SRA (PRJNA1006049).

## 2.2 Transcriptomic analysis and OR annotations

Illumina adapters were removed from the raw sequence data using *Trimmomatic* (0.39) and the trimmed reads were mapped to the Argentine ant transcriptome using *kallisto* (0.45.0) [35, 36]. The Argentine ant genome, transcriptome, and proteome were retrieved from GenBank (GCF_000217595.1). Following pseudo-alignment, all samples had between 33 million and 48 million reads mapped to the transcriptome. Transcripts that showed counts in only one sample were pruned from the dataset. Transcript counts were generated from the mapped sequence data using *DESeq2* (1.32.0), and the package was used to conduct principal components analysis and differential gene expression analysis on the transcript count data [37]. *DESeq2* identified differentially expressed transcripts by comparing the log-fold-change of transcript counts between tissues under default parameters using un-normalized count data inputs. Transcripts exhibiting expression in only a single tissue (expression-restricted transcripts) were identified by determining which transcripts had un-normalized transcript counts < 10 in other tissues.

OR proteins previously identified through hand annotation were used to BLAST against the available NCBI proteome using blastp [11, 38]. Each OR sequence was associated with the GenBank protein sequence that showed the highest percent identity match (E-value < $1 \times 10^{-40}$). Two hundred and fourteen out of 365 OR sequences were associated with GenBank protein and transcript sequences, representing around sixty percent of all ORs present within the *L. humile* genome (S1 Fig). Nine-exon ORs, thought to be involved in nestmate recognition, represent 136 of the 365 OR genes present within the *L. humile* genome. These nine-exon ORs represent most of the missing OR sequences, with 80 nine-exon ORs missing from the final OR sequences, and the remaining 18 missing sequences being other OR sequences. We proceeded our analysis with 214 ORs that we could robustly identify in the latest version of the Argentine ant transcriptome and with those we re-annotated (below).

## 2.3 Recovery and transcriptomic analysis of missing ORs

As we were unable to recover all previously annotated ORs from the NCBI data, we additionally reannotated the Argentine ant ORs. Protein sequence data for *L. humile* ORs were obtained from prior literature and each protein sequence was BLASTed against the available *L. humile* genome (GCF_000217595.1) to identify scaffold location [11]. Scaffolds were searched manually to find exon locations for each protein sequence, and a GFF file was constructed to hold the recovered OR sequences. Gene models were kept as-is based on original protein sequence data. The tissue transcriptomes were then mapped to new OR DNA sequences using *kallisto* (0.45.0) to identify ORs. Three hundred and fifty-eight out of 365 ORs were detected in transcriptome samples after reannotation. Odorant receptors were associated with their gene subfamilies based on prior literature [39]. Tandem arrays of odorant receptors were identified using the reannotated GFF file, designating a set of ORs a tandem array if they were present within the same strand and if the genomic distance between end and start sites of adjacent ORs was less than 20 kbp.

### 2.4 Phylostratigraphy

A reference sequence database was created using twenty-seven proteomes within the family Formicidae, three Hymenopteran proteomes and one Dipteran proteome (S1 Table). Each proteome was assigned a distance value based on the phylogenetic distance between the organism and the Argentine ant. The Argentine ant proteome and OR protein sequences from prior literature was compared against this database using blastp analysis (e-value threshold: $1x10^{-5}$) to identify potential orthologs for each protein in other species [11]. In instances where the queried protein found hits on multiple proteins within a single species, the queried protein was associated with the protein with the lowest e-value (i.e., most significant). Each protein in the query sequences was then associated with the node distance of the most distant species that hit on the query protein. Node distance was calculated as the number of nodes in the phylogenetic tree between *L. humile* and the query protein species.

## 3 Results

### 3.1 ORs are specifically and highly expressed in antennae

The expression patterns for each tissue were distinct from one another, and all tissue replicates clustered tightly together based on expression patterns, except the thoracic tissue which was much more loosely clustered (Fig 1A). Expression patterns of OR genes by replicate specifically show antennae tissues separating out from the remaining three tissues, which loosely cluster together (Fig 1B). We identified 4288 antennae-upregulated genes when comparing against the other tissues (p-adj < .05). A total of 376 transcripts (9%) were specifically upregulated in the antennae compared to all three other tissues (Fig 2A). A total of 72 of antennae-upregulated genes are ORs, which represents just over one third of all detected ORs present in the samples before reannotation (Figs 2B and S1); significantly more than expected by chance (Fisher's Exact Test, p < .05). After reannotation, 125 OR genes were shown to be upregulated in the antennae, representing one third of the reannotated ORs (Figs 2C and S1). All upregulated ORs were highly expressed in the antennae compared to other tissues (Figs 3 and 4). Nine-exon ORs are enriched in the antennae at a similar rate to other ORs, with 18 ORs showing enrichment compared to all other tissues, increasing to 45 after reannotation (Figs 2D, 2E and S1). Altogether, manual annotation increased the amount of ORs detected in the transcriptomic analysis from 214 to 358, with two nine-exon ORs and five other ORs from prior literature remaining missing (S1 Fig) [11]. The proportion of antennae-upregulated ORs and nine-exon ORs did not change after reannotation (Fisher's Exact Test, p > .05) (Fig 2B–2E). While other tissues show some amount of OR upregulation, only the antennae show upregulation of ORs compared to all other tissues (Fig 5). These antennae-upregulated ORs and nine-exon ORs both show high levels of enrichment compared to all other tissues (Fig 4B and 4C). Comparing OR tissue expression by OR subfamily did not show subfamily-specific expression patterns across tissues (S3 Fig). Most ORs tend to be part of a tandem array (Fig 6A). The majority of OR tandem arrays tend to be small, with few large OR arrays present in the genome (Fig 6A). However, the majority of *L. humile* ORs tend to be present in mid-sized arrays (Fig 6B). Distribution of *L. humile* ORs, nine-exon and non-, were distributed across tandem arrays proportionately to their contribution in the organism's OR repertoire (Fig 6B) ($\chi^2$, p > .05).

### 3.2 ORs are younger than the rest of the proteome

Seventy-nine percent of proteins within the Argentine ant proteome are orthologous to other proteins across Insecta (Fig 7A). OR proteins, both nine-exon and other clades, are significantly more likely to show more recent origins in Hymenoptera (58% of 9E ORs, 75% of other

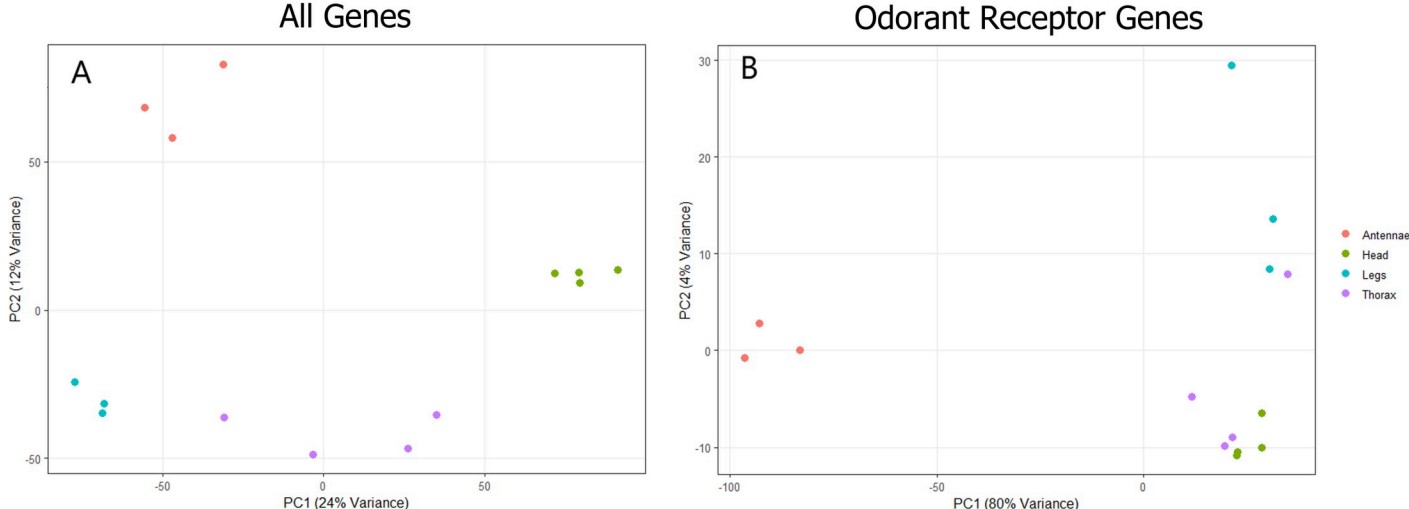

**Fig 1. Principal Components Analysis (PCA) of gene expression data.** (A) PCA of rlog-transformed gene expression data for all *L. humile* genes. Each replicate represents an mRNA extract from 200–250 workers. All tissues show clustering with each other, with thorax tissues clustering much more loosely than the other three tissues. (B) PCA of rlog-transformed gene expression data for all *L. humile* OR genes. Antennae tissue samples separate cleanly from the other three tissues, which are grouped together.

ORs), and to a lesser extent Formicidae (42% of 9E ORs, 31% of other ORs) compared to the rest of the proteome ($\chi^2$, adj. p < .05). Proteins showing increased gene expression in the antennae showed similar patterns of origin to proteins that did not show increased antennal expression ($\chi^2$, adj. p > .05). The *L. humile* proteome was found to contain 338 taxonomically restricted proteins (TRPs), although none of these proteins were found to be OR proteins according to blastp analysis. An InterProScan was conducted using *BLAST2GO* (excluding HMMTigr) and found that fifty-two TRPs were associated with DNA-binding (S2 Fig) [40, 41]. When we included all OR sequences in the phylostratigraphy analysis, the OR proteins maintained similar patterns of origin compared to the same categories in the original analysis (Fig 7B) ($\chi^2$, adj. p > .05) [11]. However, nine-exon ORs are significantly more likely to have more recent origins than other OR proteins, with 58% of nine-exon ORs showing origins as early as Formicidae, compared to only 31% of other OR proteins ($\chi^2$, adj. p < .05).

## 4 Discussion

While the overall number of genes enriched in the antennae compared to the other tissues was low [376/21797, 1.7%], a disproportionately large number [125/367, 34.0%] were identified as odorant receptors. Given the important role of antennae in chemosensation, these findings are not surprising, but they do reveal that many ORs are expressed in other body regions in addition to the antennae (Fig 4) [4]. These findings point to interesting questions on the role these ORs play in body regions not typically associated with odorant reception. Additionally, our initial analysis initially only detected 214 of the 367 ORs present in the *L. humile* genome (S1 Fig) [11]. Reannotation of the OR genes recovered most of the missing OR genes, detecting 358 out of 367 ORs in our transcriptomes (S1 Fig). This discrepancy in OR detection indicates that there are issues with the transcript annotations generated by NCBI on the *L. humile* dataset, and reannotation of the *L. humile* OR repertoire on NCBI will be necessary to correct this issue. Given the disproportionate upregulation of ORs in the antennae, the presence of ORs with seemingly no tissue-biased expression is interesting, suggesting either

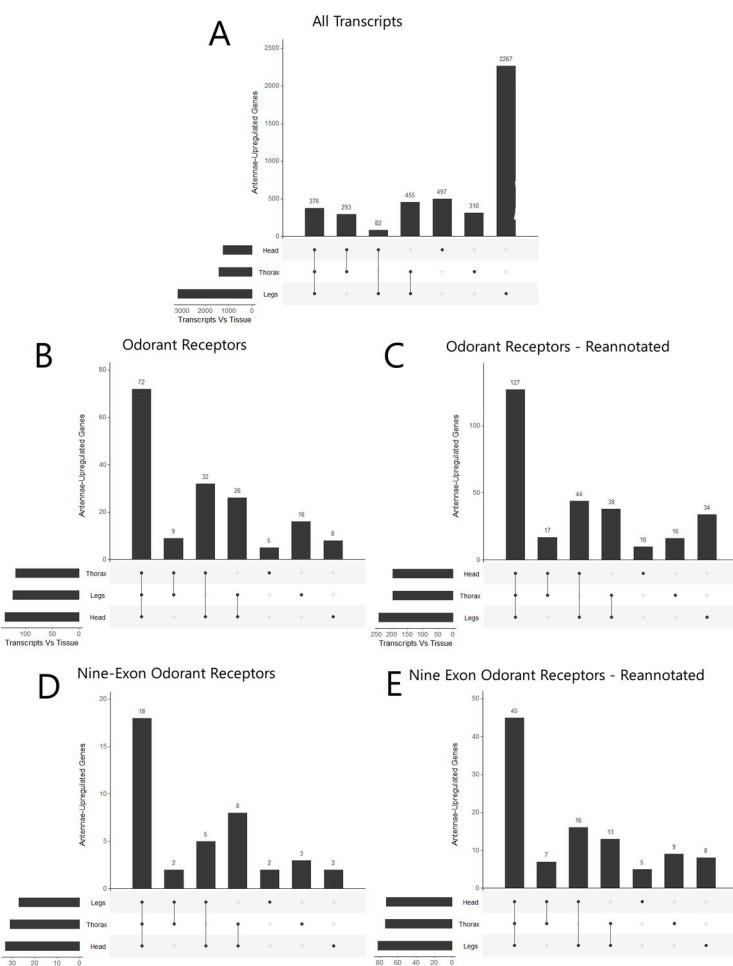

**Fig 2. Upset plots showing the number of upregulated transcripts in the antennae compared to other tissues.**
Upset plots showing antennae-upregulated transcripts versus other tissues. Vertical bars indicate the amount of transcripts upregulated in the antennae versus the compared tissues. The dots along the X-axis indicate which tissues are being compared to the antennae. The horizontal bars indicate the amount of upregulated transcripts present in the antennae versus each tissue across all comparisons. (A) Comparing all transcripts, 376 out of 21797 total genes are enriched in the antennae compared to all other tissues. (B) Comparing all OR genes, 75 total ORs out of 214 are enriched in the antennae compared to all other tissues. (C) After OR reannotation was done, 125 ORs out of 358 are upregulated in the antennae compared to all other tissues. (D) Comparing all 9-Exon OR genes, 18 9E ORs out of 56 are upregulated in the antennae compared to all other tissues. (E) After OR reannotation was done, 45 9-Exon ORs out of 134 are upregulated in the antennae compared to all other tissues.

neofunctionalization of the expanded OR gene family in *L. humile*, or odorant detection in body regions not generally associated with chemosensation (S3 Fig). ORs that do show changes in expression universally show a high level of upregulation (Figs 3 and 4), creating potential targets for RNAi-mediated knockdown of these ORs, which could prove fruitful in further investigation of how ORs interact with colony communication and nestmate recognition. Finally, when conducting transcriptomic analysis on lab colonies, it is important to keep in mind that gene expression profiles are capable of shifting as a colony is taken from the field and acclimates to lab conditions, and that results derived from lab colonies can not necessarily be extrapolated to the field [42]. However, based on prior literature none of Argentine ant's OR genes show changes in expression between lab and field colonies, so our results should remain applicable to field colonies.

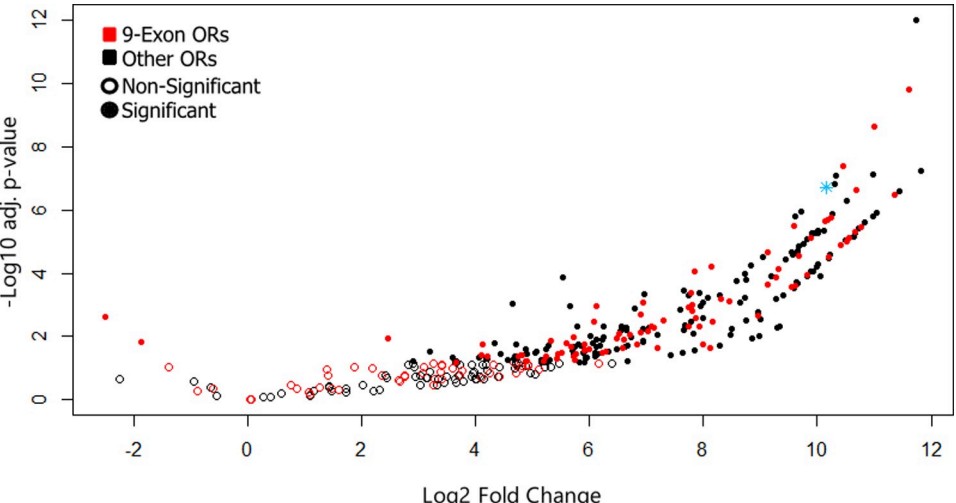

**Fig 3. Volcano plot of OR gene expression.** Volcano plot of L. humile OR antennae transcripts compared to pooled non-antennae tissues. Nine-exon ORs are represented by red dots, while other ORs are black. ORs showing significant changes in gene expression are represented by solid dots, while no change in expression are hollow dots. Most ORs showing changes in expression exhibit upregulation in antennae vs. other tissues. *Orco* is labelled in light blue.

While ORs have generally been understood to be involved in nestmate recognition in euso-cial insects, not much information is available about which ORs are involved beyond general studies indicating nine-exon ORs as likely candidates, and some studies identifying specific ORs involved in CHC detection [11, 17, 22, 23]. Our phylostratigraphy shows that these OR genes originate more recently than the rest of the *L. humile* genome, exclusively in Hymenop-tera and Formicidae. Even though 338 proteins in the Argentine ant proteome were taxonomi-cally restricted to *Linepithema humile*, none of them were ORs. This lack of any TRPs within the OR clade suggests that novel OR proteins are unnecessary in maintaining pheromone iso-lation during speciation. Additionally, the taxonomically restricted genes that were found in the *L. humile* proteome would be a worthwhile subject of research, given the lack of informa-tion associated with them. Altogether, the antennae-upregulated OR genes provide a valuable

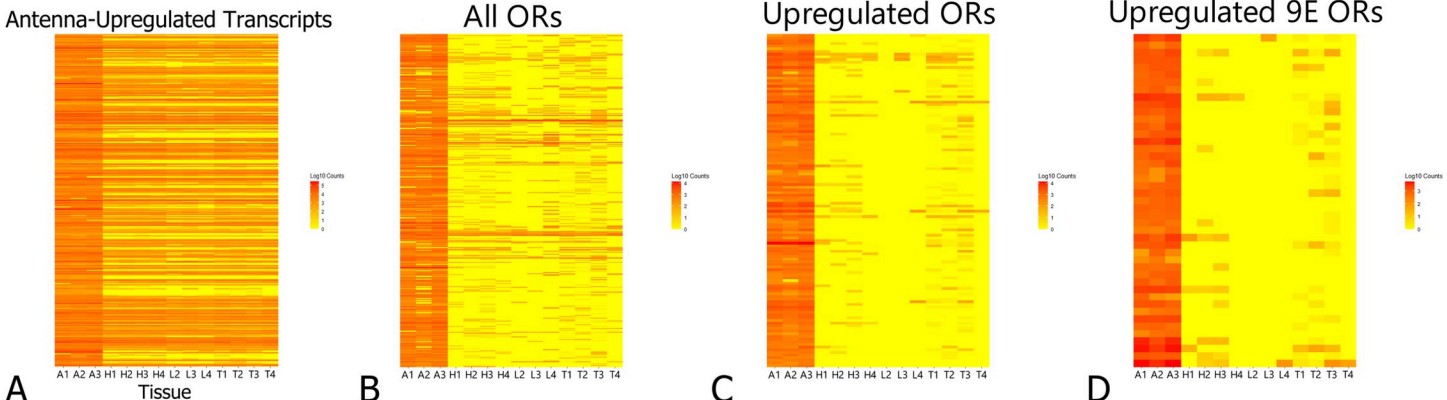

**Fig 4. Transcript count heatmaps.** Heatmaps showing Log10 transcript count data for each (A) Antennae-upregulated transcript, (B) OR, (C) antennae-enriched OR, and (D) nine-exon OR compared across all tissues using transcripts mapped to reannotated ORs. Each bar represents one transcript. Enriched ORs show high levels of enrichment compared to other tissues. Y-axis for the heatmaps can be found in S2–S5 Tables. Heatmap (A) was assembled using data mapped to NCBI reference transcriptome and is missing a portion of OR data (S1 Fig).

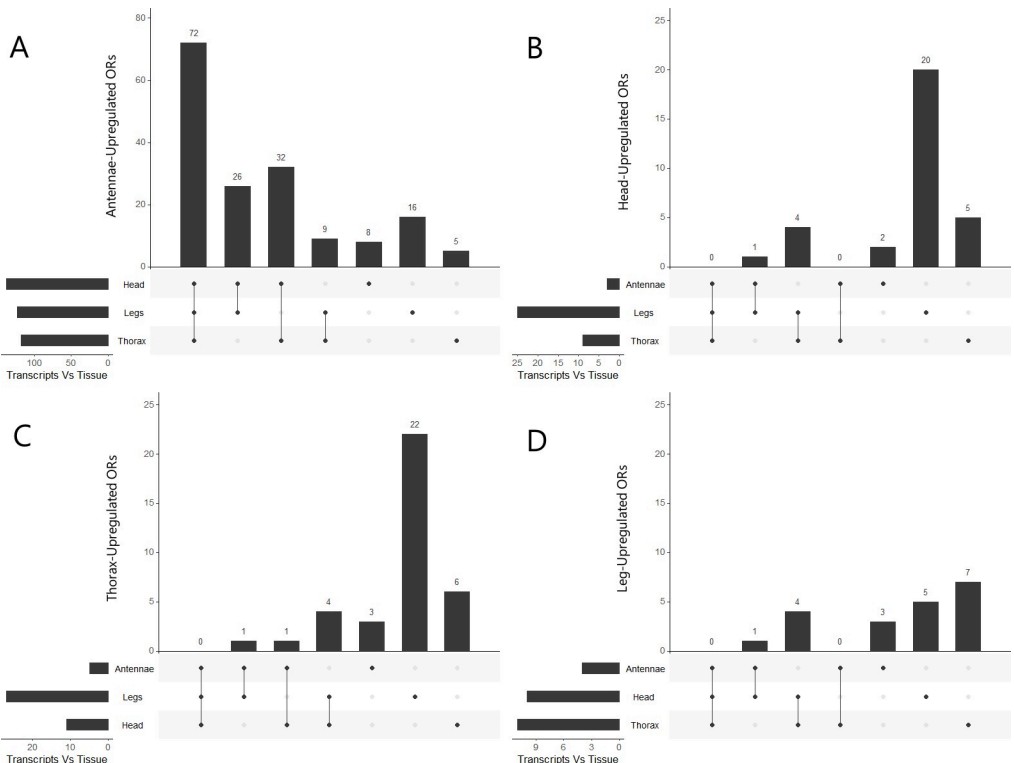

**Fig 5. Upset plots of upregulated ORs in each tissue.** Upset plots comparing OR expression in (A) Antennae, (B) Head, (C) Thorax, and (D) Legs versus the other tissues. Vertical bars indicate the amount of ORs upregulated in the tissue of interest versus the compared tissues. The dots along the X-axis indicate which tissues are being compared to the tissue of interest. The horizontal bars indicate the amount of upregulated ORs present in the tissue of interest versus each tissue across all comparisons. Only the antennae tissues show upregulation of ORs versus every other tissue. Leg tissues show lowest overall amounts of OR upregulation.

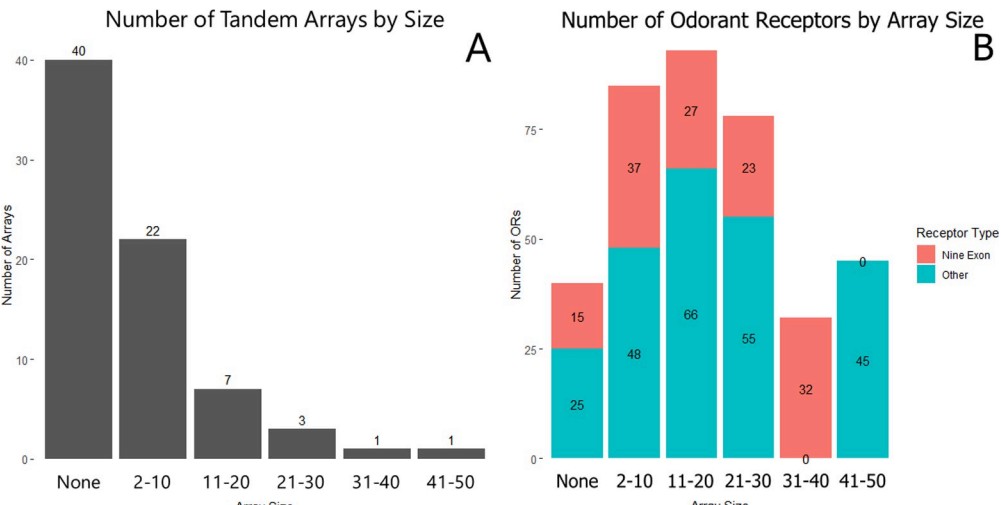

**Fig 6. Distribution of odorant receptor tandem arrays.** Histograms showing (A) the number of OR tandem arrays in the genome grouped by tandem array size, and (B) the number of ORs present in each size of tandem array. Tandem array content is inversely proportional to array size. Most ORs tend to be present in mid-size arrays, and both nine-exon and other ORs are distributed proportionately across tandem arrays.

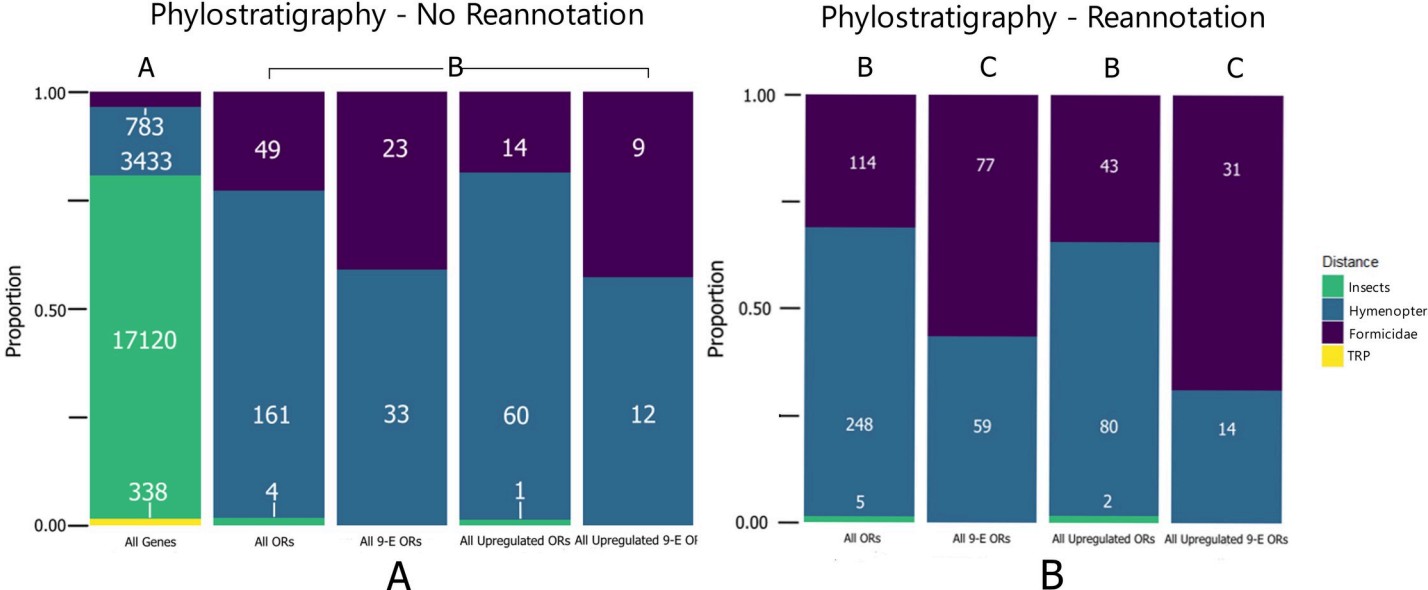

**Fig 7. Stacked barplot of phylostratigraphy node distances.** (A) Argentine ant proteome (GCF_000217595.1) against ant proteomes, using *Apis mellifera*, *Nasonia vitripennis*, and *Drosophila melanogaster* as outgroups, and (B) reannotated Argentine ant OR proteins (Smith et al. 2011). OR genes are more likely to be recently evolved, though do not contain any taxonomically restricted genes. After reannotation, 9-Exon ORs are more likely to show even more recent origins in the family Formicidae.

candidate list to further examine how Argentine ant supercolonies are maintained. Additionally, investigating the function of ORs that are primarily expressed in other tissues can provide new insights on the diversity of function by insect odorant receptors.

## Supporting information

**S1 Fig. Odorant receptors in the *L. humile* genome.** Stacked Barplot indicating the amount of expected odorant receptors (ORs) present in the Argentine ant genome and the ORs that were associated with accession numbers, split between nine-exon ORs and other ORs. Eighty 9-Exon ORs and seventy-four other ORs were unable to be matched to GenBank protein and transcript accession numbers. After OR reannotation, two 9-Exon ORs and five other ORs remain missing from the analysis.
(TIF)

**S2 Fig. InterProScan of TRPs.** Flow chart generated by BLAST2GO showing GO terms generated from InterProScan of TRPs. Fifty-two of the TRPs show protein signatures associated with DNA binding activity according to InterPro.
(TIF)

**S3 Fig. OR Heatmap by subfamily.** Heat map showing Log10 transcript counts for all *L. humile* ORs, broken up by subfamily. "NA" ORs were not included in the subfamily analysis conducted in Engsontia et al. 2015.
(TIF)

**S1 Table. Table showing genomes used for phylostratigraphic analysis.**
(XLSX)

**S2 Table. Y-axis labels for heatmap in Fig 4A.** Table containing the y-axis labels in heatmap 4A.
(XLSX)

**S3 Table. Y-axis labels for heatmap in Fig 4B.** Table containing the y-axis labels in heatmap 4B, as well as any NCBI Accession numbers associated with the genes through tBLASTn, and their % identity. Proteins with no NCBI annotations have no data associated with them.
(XLSX)

**S4 Table. Y-axis labels for heatmap in Fig 4C.** Table containing the y-axis labels in heatmap 4C, as well as any NCBI Accession numbers associated with the genes through tBLASTn, and their % identity. Proteins with no NCBI annotations have no data associated with them.
(XLSX)

**S5 Table. Y-axis labels for heatmap in Fig 4D.** Table containing the y-axis labels in heatmap 4D, as well as any NCBI Accession numbers associated with the genes through tBLASTn, and their % identity. Proteins with no NCBI annotations have no data associated with them.
(XLSX)

## Author Contributions

**Conceptualization:** Mathew A. Dittmann, Grzegorz Buczkowski.

**Data curation:** Mathew A. Dittmann.

**Formal analysis:** Mathew A. Dittmann.

**Investigation:** Mathew A. Dittmann.

**Methodology:** Mathew A. Dittmann, Grzegorz Buczkowski, Michael Scharf, Brock A. Harpur.

**Project administration:** Mathew A. Dittmann.

**Resources:** Mathew A. Dittmann, Grzegorz Buczkowski.

**Supervision:** Michael Scharf, Brock A. Harpur.

**Validation:** Mathew A. Dittmann.

**Visualization:** Mathew A. Dittmann.

**Writing – original draft:** Mathew A. Dittmann.

**Writing – review & editing:** Mathew A. Dittmann, Grzegorz Buczkowski, Michael Scharf, Brock A. Harpur.

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
