## [Decision Letter · Decision Letter 0]

25 Mar 2024

PONE-D-24-05588Comparative transcriptomics and phylostratigraphy of Argentine ant odorant receptorsPLOS ONE

Dear Dr. Dittmann,

Thank you for submitting your manuscript to PLOS ONE. After careful consideration, we feel that it has merit but does not fully meet PLOS ONE’s publication criteria as it currently stands. Therefore, we invite you to submit a revised version of the manuscript that addresses the points raised during the review process.

I agree with the reviewers that the analyses presented in this manuscript are a bit "thin". This is a dataset from four tissues, but only three of the samples are antennae - annotating the 9-exon ORs cannot have been the only goal of this study? It would be nice if the authors could follow some of the suggestions by the reviewers to add some "meat" to this manuscript in the form of further analyses.

We look forward to receiving your revised manuscript.

Kind regards,

Volker Nehring

Academic Editor

PLOS ONE

2. In the online submission form you indicate that your data is not available for proprietary reasons and have provided a contact point for accessing this data. Please note that your current contact point is a co-author on this manuscript. According to our Data Policy, the contact point must not be an author on the manuscript and must be an institutional contact, ideally not an individual. Please revise your data statement to a non-author institutional point of contact, such as a data access or ethics committee, and send this to us via return email. Please also include contact information for the third party organization, and please include the full citation of where the data can be found.

Reviewers' comments:

Reviewer's Responses to Questions

**Comments to the Author**

1. Is the manuscript technically sound, and do the data support the conclusions?

Reviewer #1: Partly

Reviewer #2: Yes

2. Has the statistical analysis been performed appropriately and rigorously? 

Reviewer #1: N/A

Reviewer #2: Yes

3. Have the authors made all data underlying the findings in their manuscript fully available?

Reviewer #1: No

Reviewer #2: Yes

4. Is the manuscript presented in an intelligible fashion and written in standard English?

Reviewer #1: Yes

Reviewer #2: Yes

5. Review Comments to the Author

Reviewer #1: Thank you for your work, it is useful results.

1. I am not sure I understand your main point. Perhaps it might be interesting to focus on your main result at first with supporting figures and then discuss the interesting parallel element in relation to the supplementary.

If the point is:

- re-annotate the OR families/9-exon

- clustering/estimate their emergence through the evolutionary history of insects

-> It might be useful to emphasise the importance of the intron phase, OR properties (etc.) in OR categorisation rather than just in the putative ecological context. Present fewer figure and discuss it under a broader paradigm than recognition and more functional/proximal.

If the point is to look at how OR distribution in this species affects recognition capabilities and explains invasive spread, then make it clearer. But I don't think this study answers that question directly so it would be care full to present it as a hypothesis rather than a result.

3. How did you handle incomplete gene models (missing regions)? What is the difference between table 2, 3 and 4 in the supplementary? They seem to be the same things. Data are missing, you may need to join the read count table and the enrichment output analysis, or a link to it.

(for more information, see the attached file)

Reviewer #2: Comments to authors

In this study, the authors have collected transcriptomic data from various tissues to reannotate the Argentine ant transcriptome and identify 9-exon odorant receptors.

While identification of these ORs is crucial for future experiments, only identifying them and analyzing their node distance from ingroup and outgroup species are not sufficient for a high-quality publication. I would therefore recommend major revisions and I hope that my comments would be helpful for revising the manuscript.

Major comments

The transcriptomic dataset needs to be analyzed further to check with ORs are specifically upregulated in different tissues. I would further recommend performing phylogenetic analyses on the ORs to investigate which subfamilies of ORs are present in antennae and other tissues. I would expect that the 9-exon subfamily to be enriched in the antennal dataset and other subfamilies to be expressed in non-antennal tissues. Additionally, the authors could extend the phylogenetic analyses with the ORs to other species of ants with well-annotated transcriptomes and this would help to identify whether there are any specific subfamilies of ORs that have undergone species-specific expansion. This could further be expanded to compare ORs in native and introduced regions to identify the ORs that might play a role in reduced aggression in the latter type.

Another potential analysis could be to look for genomic tandem arrays in the ORs and add a section on what proportion of ORs are present in tandem arrays. I do realize that long-read RNA-Seq data is a requirement for this kind of analysis, and the authors might not have that available as of now. However, I would recommend collecting long-read data and adding to this manuscript since it would improve the power of the analyses as well as the quality of the manuscript.

Specific comments

Lines 82-86: How many colonies were used to collect the workers? Were workers from different colonies treated as separate biological replicates? How many biological replicates were collected for each body part? Figure 1 shows that head and thorax have 4 replicates each, and antennae and legs have 3 replicates each. Please explain the removal of replicates for antennae and legs. Were heads and thoraces of these workers used in the analyses?

Why were the abdomen not included in the data collection? Is it definitely known that ORs are not expressed in abdomens of ants?

Lines 96-97: Were the transcript counts normalized? Please add this detail.

Line 107: Were the ORs annotated in a previous study? If so, please cite the appropriate article.

Lines 130-131: Please expand on this analysis for calculating node distance. How was the node information used in downstream analyses. Phylostratigraphy is one of the major analyses in this study and requires expansion in the methods.

A general recommendation would be to submit the analyses codes.

Lines 138-139: “We identified 4288 antennae-upregulated genes …”. Figure 2A only shows a total of 4280 antennae upregulated genes. Is this a calculation error? Are 8 genes not shown in the plot? If so, please explain why those 8 genes were dropped.

Lines 140-144: “… representing one third of the detected ORs.” Are these “detected ORs” based on the total ORs in Figure S1? It would be nice to either mention the total “detected ORs” in these two cases or refer to Figure S1.

Same comment applies for the Nine-exon ORs.

Lines 191-194: It is not clear what the authors mean by “after adding rest of the OR proteins”. Please explain.

Lines 204-206: “… odorant receptor genes do not appear to show any changes …”

It is not clear which contrasts are the authors referring to here. It sounds as if the authors are comparing ORs between lab and field colonies but not stating it clearly. Please mention if such a comparison was done. If so then the statement, “ …. so the OR findings discussed should be applicable to field colonies” is misleading.

Figure 3: Were the p-adjusted values used for this volcano plot?

Figure 5: Please expand on how these

I would recommend adding “All Genes” column to Figure 5B.

Minor comments

Lines 77, 157, 167: Italicize the species name.

Line 229 and Figure 5A: Please stick to either TRG or TRP.

6. PLOS authors have the option to publish the peer review history of their article (what does this mean?). If published, this will include your full peer review and any attached files.

Reviewer #1: No

Reviewer #2: No

---

## [Author Response · Author response to Decision Letter 0]

22 May 2024

Response: We have updated the file names for the Supplementary Tables to match the Manuscript captions.

2. In the online submission form you indicate that your data is not available for proprietary reasons and have provided a contact point for accessing this data. Please note that your current contact point is a co-author on this manuscript. According to our Data Policy, the contact point must not be an author on the manuscript and must be an institutional contact, ideally not an individual. Please revise your data statement to a non-author institutional point of contact, such as a data access or ethics committee, and send this to us via return email. Please also include contact information for the third party organization, and please include the full citation of where the data can be found.

Response: We have deposited the Phylostratigraphy results in the Purdue University Research Repository, and added a DOI to the page.

Reviewer 1:

1. I am not sure I understand your main point. Perhaps it might be interesting to focus on your main result at first with supporting figures and then discuss the interesting parallel element in relation to the supplementary.

If the point is:

- re-annotate the OR families/9-exon

- clustering/estimate their emergence through the evolutionary history of insects

-> It might be useful to emphasise the importance of the intron phase, OR properties (etc.) in OR categorisation rather than just in the putative ecological context. Present fewer figure and discuss it under a broader paradigm than recognition and more functional/proximal.

If the point is to look at how OR distribution in this species affects recognition capabilities and explains invasive spread, then make it clearer. But I don't think this study answers that question directly so it would be care full to present it as a hypothesis rather than a result.

Response: The intention of this study is to identify ORs that are likely involved in nestmate recognition by determining which nine-exon ORs are expressed primarily in the antennae. There is very little known about the role of ORs in nestmate recognition, only that nine-exon ORs are likely involved in the process. Our results show that the expression of nine-exon ORs varies across tissue, and only a subset that are selectively upregulated in the antennae are likely involved in nestmate recognition. This enables further investigation of these specific ORs to determine how they function within nestmate recognition. We have added a clarification to lines 53-59.

3. How did you handle incomplete gene models (missing regions)? What is the difference between table 2, 3 and 4 in the supplementary? They seem to be the same things. Data are missing, you may need to join the read count table and the enrichment output analysis, or a link to it.

Response: Gene models derived from protein sequence data from Smith et al. 2011 were kept as-is, we have clarified this in lines 127-128. Captions for Tables S2-5 were clarified to explain the contents of each table. Proteins with no associated data in these tables are the proteins that do not have NCBI annotations.

Reviewer 2:

The transcriptomic dataset needs to be analyzed further to check with ORs are specifically upregulated in different tissues.

Response: We have included this analysis as Figure 3, and in lines 167-168.

I would further recommend performing phylogenetic analyses on the ORs to investigate which subfamilies of ORs are present in antennae and other tissues. I would expect that the 9-exon subfamily to be enriched in the antennal dataset and other subfamilies to be expressed in non-antennal tissues. Additionally, the authors could extend the phylogenetic analyses with the ORs to other species of ants with well-annotated transcriptomes and this would help to identify whether there are any specific subfamilies of ORs that have undergone species-specific expansion.

Response: While extending the phylogenetic investigation of ant ORs is an important area of research, investigating expansion of OR subfamilies across Formicidae species has already been conducted in Engsontia et al. 2015. However, the suggestion of adding the L. humile subfamily data into our OR expression data was valuable, and we have incorporated these into Figure S3.

This could further be expanded to compare ORs in native and introduced regions to identify the ORs that might play a role in reduced aggression in the latter type.

Response: While investigating the differences underlying native and introduced populations of L. humile is vital to understanding how invasive supercolonies are formed and maintained, we believe that comparing ORs between native and introduced subfamilies will not be fruitful, given the relatively recent population separation in evolutionary time.

Another potential analysis could be to look for genomic tandem arrays in the ORs and add a section on what proportion of ORs are present in tandem arrays.

Response: We have included an analysis of ORs in tandem arrays as Figure 6, and in lines 170-174.

I do realize that long-read RNA-Seq data is a requirement for this kind of analysis, and the authors might not have that available as of now. However, I would recommend collecting long-read data and adding to this manuscript since it would improve the power of the analyses as well as the quality of the manuscript.

Response: While long-read sequencing can be valuable for investigating genomic structure, we are not able to perform long-read sequencing at this time, though it would be a fruitful avenue for future work in this area.

Lines 82-86: How many colonies were used to collect the workers? Were workers from different colonies treated as separate biological replicates? How many biological replicates were collected for each body part? Figure 1 shows that head and thorax have 4 replicates each, and antennae and legs have 3 replicates each. Please explain the removal of replicates for antennae and legs. Were heads and thoraces of these workers used in the analyses?

Response: Added lines 91-92 to explain the loss of antennal and leg samples. Head and thoracic tissues were still used for transcriptomic analysis.

Why were the abdomen not included in the data collection? Is it definitely known that ORs are not expressed in abdomens of ants?

Response: Abdomen data were not collected in this analysis due to concerns that enzymes contained in the abdomen would degrade mRNA and render any transcriptomic analysis poor-quality and wasteful.

Lines 96-97: Were the transcript counts normalized? Please add this detail.

Response: DESeq2 uses un-normalized count data as input. This information was added in line 107.

Line 107: Were the ORs annotated in a previous study? If so, please cite the appropriate article.

Response: OR annotations were obtained from L. humile draft genome paper. This paper is on line 125 in our manuscript.

Lines 130-131: Please expand on this analysis for calculating node distance. How was the node information used in downstream analyses. Phylostratigraphy is one of the major analyses in this study and requires expansion in the methods.

A general recommendation would be to submit the analyses codes.

Response: Explanation of node distance calculation was added in lines 145-146.

Lines 138-139: “We identified 4288 antennae-upregulated genes …”. Figure 2A only shows a total of 4280 antennae upregulated genes. Is this a calculation error? Are 8 genes not shown in the plot? If so, please explain why those 8 genes were dropped.

Response: This was a typographical error that has been corrected.

Lines 140-144: “… representing one third of the detected ORs.” Are these “detected ORs” based on the total ORs in Figure S1? It would be nice to either mention the total “detected ORs” in these two cases or refer to Figure S1.

Response: References to Figure S1 were added to the relevant lines (157-165).

Lines 191-194: It is not clear what the authors mean by “after adding rest of the OR proteins”. Please explain.

Response: Line 221 was rewritten to clarify that the analysis was run on reannotated ORs.

Lines 204-206: “… odorant receptor genes do not appear to show any changes …”

It is not clear which contrasts are the authors referring to here. It sounds as if the authors are comparing ORs between lab and field colonies but not stating it clearly. Please mention if such a comparison was done. If so then the statement, “ …. so the OR findings discussed should be applicable to field colonies” is misleading.

Response: Our point is that prior literature indicates that some genes change their expression between lab and field colonies of L. humile, and it’s necessary to verify whether or not ORs exhibit this behavior before our results can be extrapolated onto field conditions. We have moved this section to lines 251-257 and clarified our intentions.

Figure 3: Were the p-adjusted values used for this volcano plot?

Response: Updated Fig. 3 to clarify adj. p-value was used for volcano plot

Figure 5: Please expand on how these

I would recommend adding “All Genes” column to Figure 5B.

Response: We have included All Antenna-Upregulated Transcripts as Figure 5A.

Lines 77, 157, 167: Italicize the species name.

Response: Corrected.

Line 229 and Figure 5A: Please stick to either TRG or TRP.

Response: Corrected mentions of “TRG” to “TRP”.

---

## [Decision Letter · Decision Letter 1]

26 Jun 2024

PONE-D-24-05588R1Comparative transcriptomics and phylostratigraphy of Argentine ant odorant receptorsPLOS ONE

Dear Dr. Dittmann,

Thank you for submitting your manuscript to PLOS ONE. After careful consideration, we feel that it has merit but does not fully meet PLOS ONE’s publication criteria as it currently stands. Therefore, we invite you to submit a revised version of the manuscript that addresses the points raised during the review process. The reviewers were happy with the revised version and I like it as well. I have one minor thing that I didn't catch before: I initially stumble when people show me upset plots because they appear to lack a description of what's on the x-axis (I know it's there but it's odd not so see words), and some readers may have the same problem. It would be nice if you could add 1-2 sentences to the figure legend explaining how to read the plot, i.e. what all the bars (also the horizontal ones in the bottom left) show (upregulated in the antenna vs. all those tissues marked with a dot), and what test that was exactly (differentially expressed in antenna vs. pooled (leg, thorax, head) OR DE in all three comparisons A vs L, A vs T, A vs H separately).

We look forward to receiving your revised manuscript.

Kind regards,

Volker Nehring

Academic Editor

PLOS ONE

Journal Requirements:

Reviewers' comments:

Reviewer's Responses to Questions

**Comments to the Author**

1. If the authors have adequately addressed your comments raised in a previous round of review and you feel that this manuscript is now acceptable for publication, you may indicate that here to bypass the “Comments to the Author” section, enter your conflict of interest statement in the “Confidential to Editor” section, and submit your "Accept" recommendation.

Reviewer #1: All comments have been addressed

Reviewer #2: All comments have been addressed

2. Is the manuscript technically sound, and do the data support the conclusions?

Reviewer #1: Yes

Reviewer #2: Yes

3. Has the statistical analysis been performed appropriately and rigorously? 

Reviewer #1: Yes

Reviewer #2: Yes

4. Have the authors made all data underlying the findings in their manuscript fully available?

Reviewer #1: Yes

Reviewer #2: Yes

5. Is the manuscript presented in an intelligible fashion and written in standard English?

Reviewer #1: Yes

Reviewer #2: Yes

6. Review Comments to the Author

Reviewer #1: I am still struggling with the correlation between ORs distribution and invasive species concepts. "These ORs are crucial for maintaining colony cohesion that allows invasive ant species to dominate colonized environments." sentence from abstract. For the rest, the expections were fulfilled. Thank you for your work.

Reviewer #2: (No Response)

7. PLOS authors have the option to publish the peer review history of their article (what does this mean?). If published, this will include your full peer review and any attached files.

Reviewer #1: **Yes: **Melanie Bey

Reviewer #2: No

---

## [Author Response · Author response to Decision Letter 1]

29 Jun 2024

Response: We have updated captions for Figures 2 and 3 to include an explanation of how to read the upset plots.

---

## [Editor Report · Decision Letter 2]

9 Jul 2024

Comparative transcriptomics and phylostratigraphy of Argentine ant odorant receptors

PONE-D-24-05588R2

Dear Dr. Dittmann,

We’re pleased to inform you that your manuscript has been judged scientifically suitable for publication and will be formally accepted for publication once it meets all outstanding technical requirements.

Kind regards,

Volker Nehring

Academic Editor

PLOS ONE
---

## [Editor Report · Acceptance letter]

19 Jul 2024

PONE-D-24-05588R2 

PLOS ONE

Dear Dr. Dittmann, 

I'm pleased to inform you that your manuscript has been deemed suitable for publication in PLOS ONE. Congratulations! Your manuscript is now being handed over to our production team.

Kind regards, 

on behalf of

Dr. Volker Nehring 

Academic Editor

PLOS ONE